



# Storm-induced sediment supply to coastal dunes on sand flats

Filipe Galiforni-Silva[1], Kathelijne M. Wijnberg[1], and Suzanne J.M.H. Hulscher[1]

[1]Water Engineering & Management, Faculty of Engineering Technology, University of Twente, P.O. Box 217, 7500 AE, Enschede, The Netherlands

**Correspondence:** Filipe Galiforni-Silva (f.galifornisilva@utwente.nl)

**Abstract.** Growth of coastal dunes requires a marine supply of sediment. Processes that control the sediment transfer between the sub-tidal and the supra-tidal zone are not fully understood, especially in sand flats close to inlets. It is hypothesised that storm surge events induce sediment deposition on sand flats, providing fresh material for aeolian transport and dune growth. The objective of this study is to identify which processes cause deposition on the sand flat during storm surge conditions and discuss the relationship between the supra-tidal deposition and sediment supply to the dunes. We use the island of Texel as a case study, of which multi-annual topographic and hydrographic data sets are available. Additionally, we use the numerical model XBeach to simulate the most frequent storm surge events for the area. Results show that supra-tidal shore-parallel deposition of sand occurs in both the numerical model and the topographic data. The amount of sand deposition is directly proportional to surge level and can account for more than a quarter of the volume deposited at the dunes yearly. Furthermore, storm surges are also capable of remobilising the top layer of sediment of the sand flat, making fresh sediment available for aeolian transport. Therefore, in a sand flat setting, storm surges have the potential of reworking significant amounts of sand for aeolian transport in periods after the storm, and as such can also play a constructive role in coastal dune development.

## 1 Introduction

Coastal dunes are important natural flood defence features. They grow at the interface between land and sea by the interaction of biological processes, physical processes and geological conditioners (Hesp, 1983; Sherman and Bauer, 1993; Hesp, 2002; Bauer and Davidson-Arnott, 2002; Hesp and Walker, 2013; Delgado-Fernandez and Davidson-Arnott, 2011; van Puijenbroek et al., 2017). Key requirements for the development of coastal dunes are the availability of sediment, space for dune growth, suitable climate conditions (e.g. wind, waves, vegetation, rain) and time for its development (Hesp, 1983, 2002; Bochev-van der Burgh et al., 2009; Bauer et al., 2009; Bochev-van der Burgh et al., 2011; Keijsers et al., 2015; van Puijenbroek et al., 2017; Silva et al., 2018, 2019).

The amount of available sediment controls aspects like dune type and morphology, vegetation growth and growth rate (Eastwood et al., 2011; Hesp, 2002; Short and Hesp, 1982; Houser, 2009). The sea is the primary source of sediment for coastal dunes. Wave-driven currents, oscillatory components of the incident wave motions and effects of infra-gravity waves on currents are responsible for transporting sediment onshore, leading to a continuous supply of sediment from the sub-tidal (i.e. zone that stays below sea level) to the intertidal zone and, consequently, sub-aerial zone (i.e. also known as supra-tidal zone, refers to the zone above the high tide level) (Aagaard, 2014). Aagaard et al. (2004) link the occurrence of onshore bar



migration and its subsequent welding to the coast with sediment supply towards the dunes. Anthony et al. (2006) show that, for a tide-dominated beach in the coast of France, annual dune accretion depends on bar welding phenomena related to storm processes, which could account for 48% of the overall dune change. Anthony (2013) shows that for the southern North Sea coastal system (i.e. French and Belgium coast), the highest rate of foredune accretion is associated with areas where underwater

sandbanks have migrated onshore in the past century, thus leading to an increased supply condition for the dunes.

Most studies on beach-dune systems and sediment transfer between sub-tidal and supra-tidal zones only consider locations away from inlets (Anthony et al., 2006; Anthony, 2013; Aagaard et al., 2004; Reichmüth and Anthony, 2007). Inlet-driven processes such as shoal attachment and channel migration can drive changes in the adjacent coastlines (Fitzgerald et al., 1984; Fenster and Dolan, 1996; Robin et al., 2009; Elias and Van Der Spek, 2006), which in turn can influence sub-tidal/sub-aerial

sediment exchange and coastal dune behaviour (Ruessink and Jeuken, 2002; Aagaard et al., 2004; Anthony et al., 2006; Cohn et al., 2017). For barrier islands in the Dutch Wadden sea region, beaches close to inlets commonly developed as sand flats due to long-term morphodynamics of its ebb-tidal delta systems, as illustrated by De Hors sand flat at Texel island (The Netherlands) (van Heteren et al., 2006; Elias and Van Der Spek, 2006). Those sand flats are large (scale of km) and present great potential for dune growth due to their large beach width, wind velocities, and climate (Bauer et al., 2009; Houser and

Ellis, 2013). A recent analysis of annual topographic data (Wijnberg et al., 2017) suggested that supra-tidal storm deposits may form a source for sand supply towards the dunes. However, it is unclear during which conditions supra-tidal deposition occurs and whether the amount deposited can be considered significant for dune growth and development.

Therefore, the objective of this study is to identify processes and storm conditions that cause deposition on the sand flat during storm surge flooding and to discuss the relationship between the supra-tidal deposition and sand supply to the dunes.

We use a site in the Netherlands (Texel island) as a case study, for which we analysed multi-annual topographic data sets together with a field survey and the application of a numerical model to investigate bed level changes at the sand flat during storm surge flooding events.

The paper outline is as follows: Section 2 describes the study area characteristics; Section 3 presents the available data, its treatment and usage and explains the numerical model, its concepts, initial conditions and assumptions. Section 4 presents the

results starting with multi-annual topographic data analysis, the field survey and finally results from the numerical simulations. The paper closes with a discussion section (5) and conclusion (6).

## 2   Study area

On the southern side of Texel island (The Netherlands), bordering the Marsdiep Inlet, long-term ebb-tidal dynamics built a sand flat (named "De Hors") where dunes have been emerging, at least over the past 20 years (Figure 1). The flat has an

approximate area of 3 $km^2$. According to Silva et al. (2018), around 1.2 $10^6$ $m^3$ of sand has been deposited in the dunes between 1997 and 2015. Furthermore, the dune area at the Hors can be separated into three distinct zones: a western part, more exposed to wave action, a central zone, and an eastern part, which receives less wave action. According to Silva et al. (2018), the western zone accounted for 60% of the total dune volume increase, which emerged mostly as a linear dune ridge, similar to





the foredunes found along the Dutch coast away from inlets. The central dune zone accounted for about 30% of the total dune volume increase and emerged as coppice-like dunes. The eastern zone presented the lowest dune volume increase and evolved as a linear dune ridge.

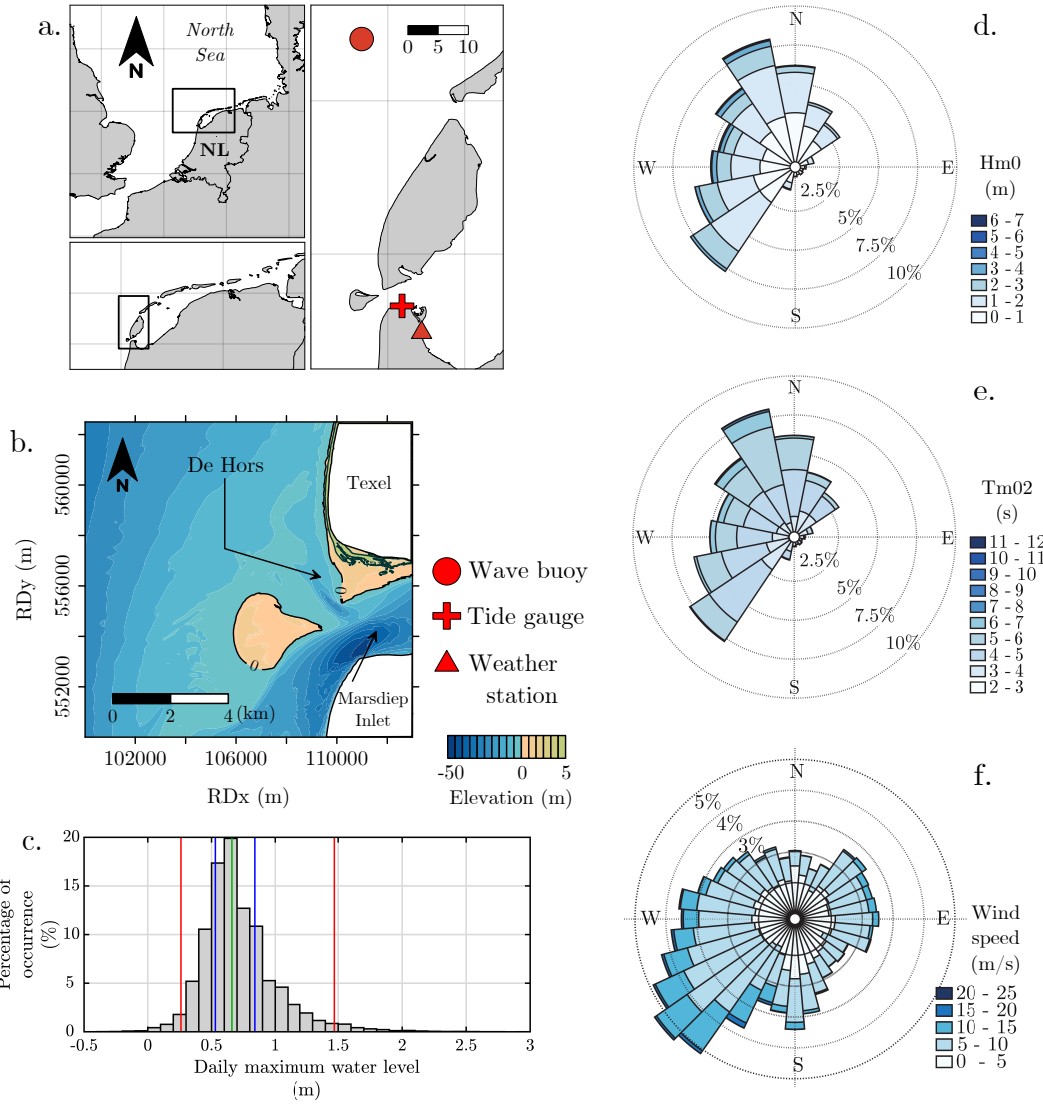

**Figure 1.** Study area of Texel. (a.) Geographic location. Red symbols represent the locations of the wave buoy (circle), weather station (triangle) and tide gauge (cross) used in this paper. (b.) Topographic setting highlighting the sand flat used as a case study. (c.) Histogram of daily maximum water levels, where blue lines represent the 25 and 75 quartiles, and the red lines 2.5 and 97.5 percentiles. Overall characteristics of the area for wave and wind climate are shown by directional histograms for $Hm0$ (d.), $Tm02$ (e.) and Hourly averaged wind speed (f.)



The Marsdiep inlet is classified as a mixed-energy wave dominated inlet, with a gorge width of 3 km and a channel depth up to 50 meter (Elias and Spek, 2017). The inlet has an asymmetric ebb-tidal delta that is mostly conditioned by side-effects of a past large-scale engineering project (i.e. construction of the "Afsluitdijk") (Elias and Van Der Spek, 2006; Elias and Spek, 2017). The sand flat is exposed to wind most of the time and is flooded only during storm surge events. The main wind direction
is from the southwest, whereas waves predominantly come from southwest and northwest directions (Figure 1). The mean tidal range is 1.41 meters, with a mean spring high tide level (MSHTL) of 0.66 meters (Rijkswaterstaat, 2013).

In the present paper, a storm surge is defined by its maximum water level following the classification used by the Dutch Ministry of Infrastructure and Water Management ('Rijkswaterstaat"). Storms surges with maximum water levels between 75 quartile and 1.9 meters above MSL are classified as mild, whereas maximum water levels between 1.9 and 2.6 meter above
MSL are classified as normal storms surges, and above 2.6 meters are classified as an extreme storm surges. To determine the local storm climate, we used an approximately 29-year long time series of hourly water levels collected at the 'Den Helder' tide gauge in the channel margin together with hourly wave data from a wave buoy for the same period (Figure 1). Daily maximum water levels were extracted from the time series and used as a proxy for storms. Results show that 73.31% of the daily maximum water levels lie below the 75 quartile level (0.86 meters), whereas 26.15% can be considered mild storms.
From the mild storms, 22.26% lies between MSHTL and the 97.5 percentile (1.32 meters), with only 3.88% representing water levels above the 97.5 percentile. Only 0.54% is classified as storms (0.5%) or extreme storms (0.04%).

Wave conditions related to storms are separately classified for each storm surge class (i.e. mild, storm and extreme storm) in Table 1. For mild storms, waves come from three directions (SW, W and NW), with relatively similar occurrence (25.8%, 30.1% and 23.4%). For storms and extreme storms, waves tend to come from more northerly directions.

**3   Numerical modelling and data analysis**

To achieve our objectives, we follow two main approaches: numerical modelling and analysis of observational data. The analysis of multi-annual topographic surveys focuses on the occurrence of deposition above the MSHTL, as well as the overall erosion/deposition patterns in the flat. This analysis is complemented by field observations collected after a storm surge event to qualitatively identify the effects of a single storm onto the sand flat. The numerical modelling is used to analyse in depth
which processes control the deposition on the sand flat during storm surge flooding and identify which storm conditions will lead to sand deposition.

**3.1   Data and Field Campaign**

To analyse beach-dune behaviour over the sand flat area, we used annual LiDAR data from 1997 up to 2018 provided by the Dutch Ministry of Infrastructure and Water Management ('Rijkswaterstaat"). Survey dates vary over the years, with a tendency
of flights being done after the most energetic period (Figure 2). The data is available at a horizontal resolution of 5 meters up to 2013, when a finer horizontal resolution of 2 meters became available. The vertical accuracy is within 0.08 meters.





**Table 1.** Characteristics of the local storm surge climate based on wave buoy measurements. Occurrence relates to the percentage of occurrence of storms with those characteristics over the population of mild, storm or extreme storms.

| Storm Surge classification | Wave Direction | $Hm0$ (m) | $Tm02$ (s) | Occurrence (%) |
|---|---|---|---|---|
| Mild | SW | 2 - 3 | 5 - 6 | 17.3 |
| | SW | 3 - 4 | 5 - 6 | 8.5 |
| | W | 2 - 3 | 5 - 6 | 5.2 |
| | W | 3 - 4 | 5 - 6 | 16.7 |
| | W | 4 - 5 | 5 - 6 | 8.2 |
| | NW | 3 - 4 | 6 -7 | 8.8 |
| | NW | 4 - 5 | 6 -7 | 14.6 |
| Normal | SW | 3 - 4 | 6 -7 | 9.1 |
| | W | 2 - 3 | 5 - 6 | 6.8 |
| | W | 3 - 4 | 6 - 7 | 25.0 |
| | W | 4 - 5 | 6 - 7 | 15.9 |
| | NW | 3 - 4 | 6 - 7 | 15.9 |
| | NW | 4 - 5 | 6 - 7 | 11.4 |
| Extreme | W | 4 - 5 | 6 - 7 | 33.3 |
| | NW | 5 - 6 | 7 - 8 | 33.3 |
| | NW | 5 - 6 | 8 - 9 | 33.3 |

From the LiDAR data, we calculated annual changes in elevation and volume of the dune field and the sand flat. The dune area has been defined by the 3-m contour, whereas the sand flat area has been defined as the area between 1.5 meters and the MSHTL. The dunefoot level along the Dutch coast is widely assumed to be approximately 3 meters + NAP (i.e. *Normaal Amsterdams Peil* - in Dutch - the Dutch reference level, which is close to the mean sea level) (de Winter et al., 2015; Ruessink and Jeuken, 2002; Quartel et al., 2008; Keijsers et al., 2014; de Vries et al., 2012; Silva et al., 2019; Duarte-Campos et al., 2018; Donker et al., 2018). The value is based on past measurements in which dunefoot was defined as a visible break in slope between beach and dune and was roughly 3 meters + NAP (de Vries et al., 2012). Furthermore, we use variance maps to analyse the stability of the sand flat using the entire topographic data set. Variance maps show the elevation variance at each grid node, which highlights areas on which elevation changes occurred in a larger magnitude. Furthermore, elevation difference maps have been used to define erosion and accretion trends between subsequent surveys. Difference maps are calculated by subtracting the elevation survey of the previous year from the next year, such that positive values relate to accretion and negative to erosion. Thus, areas with low growth trend and high variance values suggest that even though no accretion/erosive trend occurs, the elevation varies considerably in time. Moreover, to determine whether a location presented more accretive or erosive events in time, we built maps of occurrence of accretion and erosion events (i.e. number of times which a particular location had experienced an annual bed level change larger than ±0.16 meters). Thus, areas with increased frequency of accretive/erosive





events are highlighted. Therefore, the trend map will reveal the overall growth trend and spatial variations therein, variance map will show the variability in bed level, without consideration of a temporal structure, and the occurrence map will show areas where more accretive or erosive events occurred, regardless of its magnitude and trends over time.

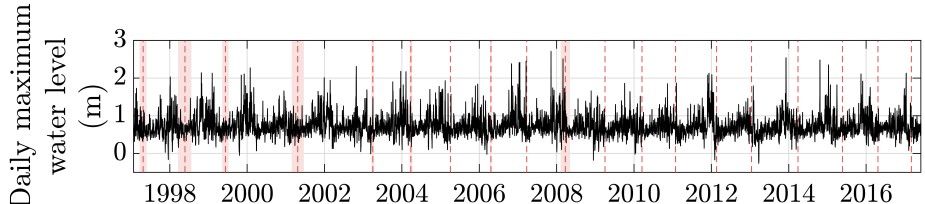

**Figure 2.** Daily maximum water level time-series, including indications of the periods when the topographic surveys were executed. Dashed red lines represent the exact date used in the analysis, whereas the pink-shadowed region represent the potential period during which the measurements took place (only for surveys where the exact date were not available).

To analyse the effects of a storm surge on the surface layer of sediment at the sand flat, we executed a field survey before and
5  after an event that flooded the sand flat of the Hors on January, 2017. Elevation data was acquired at six locations and along a transect across the sand flat to check whether changes in elevation occurred and, if so, in which order of magnitude (Figure 3). Elevation data was acquired using an RTK-DGPS system.

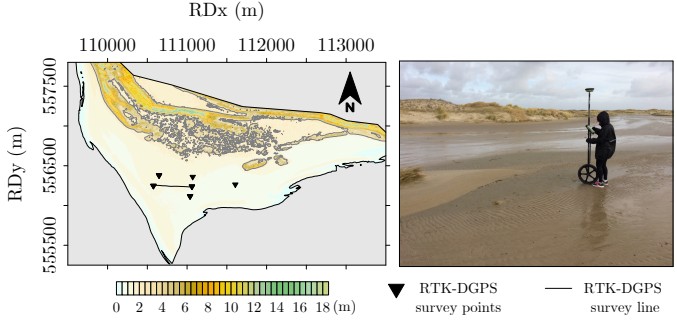

**Figure 3.** Location of the RTK-DGPS survey transects and points.

## 3.2 The XBeach Model

Considering that the topography is available on an annual basis (therefore, no data on the impacts of a single storm surge event
10  is available), we choose to simulate the accretion/erosion patterns onto the sand flat induced by the most frequent storm surge conditions in the area. The main goal is to identify, in an event scale, during which storm conditions deposition onto the sand flat occurs and understand the leading hydrodynamic processes underlying such deposition.





### 3.2.1 Model Structure

The XBeach model (Roelvink et al., 2009) is a process-based model developed to simulate hydrodynamic and morphodynamic processes on sandy coasts. It has been developed to work on a time-scale of storms and for coastal stretches of the order of kilometres in length. The model solves the 2D horizontal shallow water equations, including capabilities of time-varying

wave action balance, roller energy balance, advection-diffusion equation, sediment transport and bottom change (Elsayed and Oumeraci, 2017; Roelvink et al., 2009; Deltares, 2018). The model includes the hydrodynamic processes of short-wave transformation (refraction, shoaling and breaking), long wave transformation, wave-induced setup and unsteady currents, as well as overwash and inundation. The morphodynamic processes include bed load and suspended sediment transport, dune face avalanching, bed update and dune breaching (Deltares, 2018). The main difference of the XBeach model compared to other

process-based models for coastal areas is the capability of including the effects of infragravity waves through solving long-wave motions created by time-dependent cross-shore wave height gradients (Roelvink et al., 2009). For the present study, we run the model in surfbeat mode, where the short wave variations on the wave group scale and the long waves associated with them are resolved (Deltares, 2018). The model has been extensively validated and applied in a range of coastal settings, including overwash and storm surge flooding (de Vries, 2009; Roelvink et al., 2009; McCall et al., 2010; Elsayed and Oumeraci, 2017;

Vet, 2014; Nederhoff, 2014; Engelstad et al., 2017). Detailed information on model formulation and validation can be found in Roelvink et al. (2009) and Deltares (2018).

### 3.3 Scenarios

Based on the local storm climatology (Table 1), we selected 12 actual storm surge events that occurred between 1990 and 2017 to represent the most frequently occurring storm conditions in each of the three storm surge categories. Choices have been made

to ensure that we simulated at least one storm surge from each wave direction represented. For the domain, we used LiDAR and bathymetric data available for the year 2009, the same year of the storm chosen for validation. Thus, only hydrodynamic boundary conditions are different for each scenario. Bathymetric data is available at a 20x20 meter grid, with vertical accuracy between 0.11-0.4 meters, whereas topographic LiDAR data is available at a 5x5 meter grid, with vertical accuracy within 0.08 meters. For each storm, its wave characteristics have been gathered from data available from a nearby wave buoy (Figure 1).

Simulated scenarios are shown in Table 2.

From the simulations, we relate bed level change on the flat with local hydrodynamic characteristics (i.e. $Hrms$, $u$ and $v$ gradients), in order to identify which driving force would explain most of the bed level change. $u$ and $v$ gradients refer to the zonal and meridional components of the local depth-averaged flow velocity. We do this by analysing how the morphology and hydrodynamics evolve in time, at a location where deposition occurs and following the time-series of bed level change and

hydrodynamic processes. Furthermore, to study whether storm strength influences the amount of deposited volume onto the sand flat, we correlate final sand volumes deposited with imposed storm characteristics (i.e. Maximum water level imposed at the boundary, $Hm0$, wave direction and $Tp$). Violin plots are used to identify on which elevation the volume changes occurred. Violin plots are essentially box plots with the addition of a rotated probability density plot on each side.





**Table 2.** Characteristics of the simulated scenarios. Deposited volume refers to the amount of sand deposited onto the sand flat according to the simulation results.

| Scenario | Date | Duration simulated (hours) | $Hm0(m)$ | $Dir(deg.)$ | $Tp(s)$ | Max. Water Level (m) | Deposited volume ($m^3$) | Storm class |
|---|---|---|---|---|---|---|---|---|
| a. | 25-26/10/2005 | 8 | 3.0 | 235 | 5.6 | 1.7 | 474 | Mild |
| b. | 1/10/2008 | 7 | 3.0 | 258 | 5.3 | 1.7 | 1198 | Mild |
| c. | 29/10/2017 | 8 | 3.8 | 310 | 6.4 | 1.7 | 7084 | Mild |
| d. | 23/11/2009 | 8 | 2.9 | 253 | 5.4 | 1.9 | 7339 | Mild |
| e. | 04/10/2009 | 8 | 3.8 | 297 | 6.3 | 2.1 | 14680 | Storm |
| f. | 25/10/1998 | 8 | 4.0 | 292 | 6.5 | 2.4 | 15322 | Storm |
| g. | 21/12/2003 | 9 | 5.8 | 350 | 8.1 | 2.5 | 26958 | Storm |
| h. | 27/10/2002 | 9 | 3.5 | 247 | 6.2 | 2.6 | 10717 | Storm |
| i. | 30/01/2000 | 9 | 3.6 | 298 | 6.7 | 2.6 | 16173 | Storm |
| j. | 22/10/2014 | 9 | 4.7 | 323 | 7.1 | 2.8 | 19363 | Extreme Storm |
| k | 09/11/2007 | 10 | 5.8 | 337 | 8.1 | 3.0 | 29863 | Extreme Storm |
| l. | 26/02/1990 | 10 | 5.0 | 285 | 7 | 3.2 | 18601 | Extreme Storm |

### 3.3.1 Validation

Regarding the validation of the numerical model, only one hydrodynamic dataset was available to assess model performance for the present study. The ferry that links Texel island with the mainland crosses the Marsdiep inlet every half an hour from 6 AM up to 9:30 PM with an Acoustic Doppler current profiler (ADCP) mounted, performing detailed flow measurements. The data acquired and treated by the Royal Netherlands Institute for Sea Research (NIOZ) has been made available for the year of 2009 by Duran-Matute et al. (2014). One limitation of the data set is that the ferry does not sail at night. Furthermore, the ferry also does not sail when the water level exceeds 2 meters above NAP. That would introduce gaps and limit the number of possible storms for validation to only periods of mild storms surges. To assess model performance, we use the storm of 04/10/2009, which reached maximum water levels of 1.8 meters and these levels were measured for three hours.

It is important to note that this dataset only gives information related to the inner side of the system (i.e. tide-dominated), thus being not optimal for validation given our interest being on the sand flat, where wave-driven processes are expected to play a significant role. However, given the lack of hydrodynamic data during storm surges for the area and considering that the XBeach model has been validated in a broad range of applications, we use default settings for the present study. Additionally, we used the dataset mentioned above for a limited validation check.

Validation results show reasonable agreement between measured and simulated currents (Figure 4). The best fit is associated with meridional components (i.e. perpendicular to the inlet throat), with a $R^2$ of 0.63 and RMSE of 0.17. For the zonal components (i.e. parallel to the inlet throat), the model underestimated values, especially when the flow presented high velocities, with $R^2$ of 0.51 and RMSE of 0.41





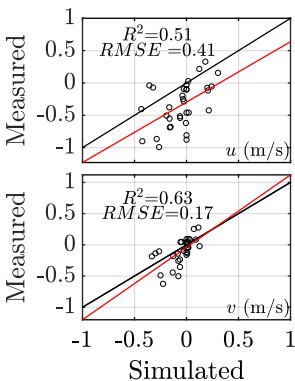

**Figure 4.** Validation results. Comparison of zonal and meridional components of the depth-averaged flow between simulated and measured data. The location of each point is defined by the ferry location at the time of the measurement and paired with the simulation accordingly. R-squared and RMSE are displayed for in the upper part of each scatter plot. Black line is a diagonal reference line (i.e. from (-1,-1) to (1,1), whilst red line represent the least-squares lines.)

## 4 Results

### 4.1 Supra-tidal development

Figure 5 presents elevation difference maps between consecutive years. Maps show that deposition patterns in the supra-tidal zone occur between at least ten different years. For some years like between 1998-1999 (b.) and 2003-2004 (e.), the deposition happens to extend from the north to the south of the flat, and it happens at least 100 meters landward of the mean spring high tide level (i.e. higher elevations). For other periods like (j., l. and r.), the deposition happens much closer to the MSHTL, although also oriented from north to south. For others, the deposition occurs only in the southern tip of the flat, like m. Erosion patterns exceeding 0.16 meters occur only between a few years, and mostly at locations close to the MSHTL. When looking at the map of accretion/erosion occurrence (balance of occurrence of accretion and erosive trends between years - Figure 6b), we can also see that a zone with more accretive than erosive years occurs in a well-formed shore-parallel shape above MSHTL. Thus, we conclude that there is a zone of sediment deposition in the west margin of the flat above mean spring high tide level.





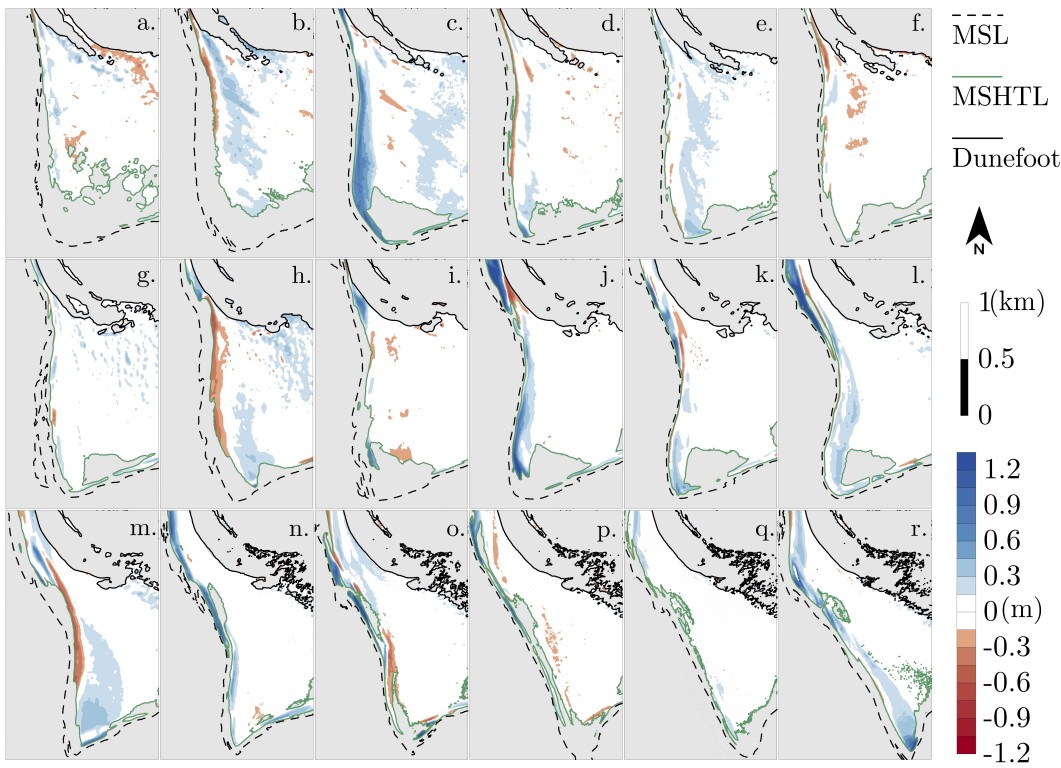

**Figure 5.** Difference maps for the periods between 1997 and 2017 focusing on the supra-tidal area. Letters refer to the specific years from which the subplots where calculated: 1998-1997 (a.), 1999-1998 (b.), 2001-1999 (c.), 2003-2001 (d.), 2004-2003 (e.), 2005-2004 (f.), 2006-2005 (g.), 2007-2006 (h.), 2008-2007 (i.), 2009-2008 (j.), 2010-2009 (k.), 2011-2010 (l.), 2012-2011 (m.), 2013-2012 (n.), 2014-2013 (o.), 2015-2014 (p.), 2016-2015 (q.) and 2017-2016 (r.). Most plots show a one-year difference, with exception of plots c. and d. which represent the difference between 1999-2001 and 2001-2003, respectively, due to the absence of surveys in the years 2000 and 2002.





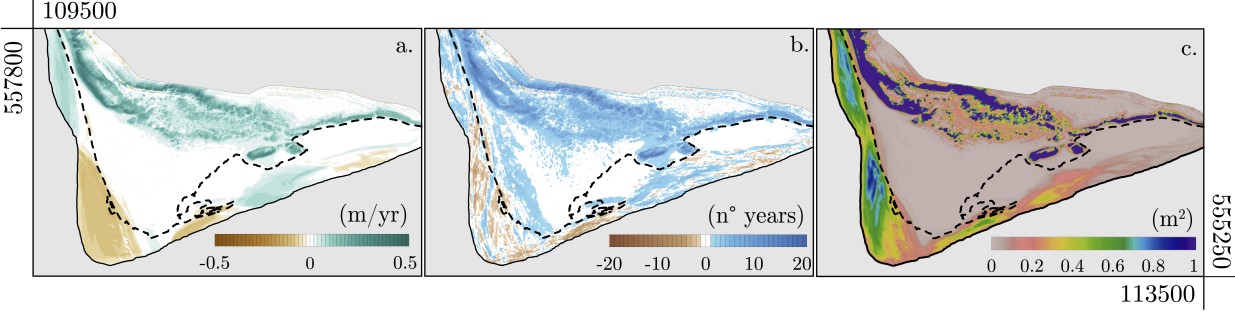

**Figure 6.** a - Average annual elevation change based on LiDAR data from 1997 up to 2017. Dashed lines show the average position of the MSHTL. b - Net number of years where accretion/erosion occurrences from the difference maps were greater than 0.16 meters. Negative values mean that the location had more erosive years than accretive, whereas positive values mean that the location had more accretive years than erosive. c - Variance of the elevation.

In terms of volume, the supra-tidal depositional zones account for values in the order of $10^4$ $m^3$, with average values of 2.5 $10^4$ ($\pm$ 1.8 $10^4$) $m^3$ over the surveyed period, and maximum numbers reaching values one order of magnitude higher (Figure 7). The deposited volume over the sand flat shows no correlation with either measured maximum water levels or median values of storm surge levels between surveys.

Even though there is a deposition zone, the flat as a whole does not present any long-term average deviation between MSHTL and the dunefoot in terms of elevation. Figure 6a shows the average year to year elevation change. In the upper part, accretion trends relate to dune growth, with elevation change up to 0.5 meters per year. Also, regions of accretion and erosion on levels below mean spring high tide level (MSHTL - dashed lines) range to values between -0.25 and 0.25, approximately. Average annual elevation change in the central part of the flat is minimal, with values within the measurement error. Variance maps

related to the elevation between each year (Figure 6c) also show that values are higher for sub-tidal zones and zones where dunes have been growing compared to the centre of the sand flat, which has variance values smaller than 0.01 for most of the zone. This suggests that not only average annual elevation changes are close to 0, but also that the area presents low temporal variability in elevation. Thus, to maintain the rate of change close to 0, erosive years must be higher in magnitude than accretive years. Moreover, the location of such deposition zone being above MSHTL means that either the deposition is caused by water

levels above mean spring high tide level or by other transport mechanisms like aeolian transport.



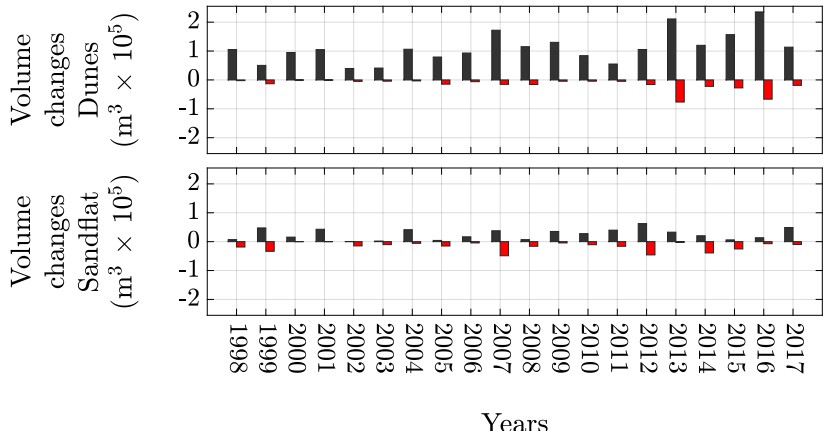

**Figure 7.** Annual volume changes for the sand flat and dune area considering only cells where elevation change was greater than 0.16 meters (approximately the maximum possible error based on the allowed error for each LiDAR survey.)

Regarding dune growth, on average $1.1\ 10^5$ ($\pm 5.2\ 10^4$) $m^3$ of sand per year is deposited on the dunes, which represents a change in the height of the dune area of 0.28 meters per year on average. Overall, a total of $2.2\ 10^6$ $m^3$ of sand has been deposited in the dune part between 1997 and 2017. This sediment resulted in an average increase in elevation of 2.51 meters and an expansion of the dune field by $9.2\ 10^5$ $m^2$. Potentially, when comparing the volume of sand deposited at the sand flat

5   and the dunes, the yearly average volume deposited on the sand flat over the years represents 27.8% of the yearly average volume change of the dunes.

Results from the field survey show that bedforms with an average height of 11 centimetres and length of 150-250 centimetres, approximately, developed on the west portion of the sand flat, gradually diminishing their size towards the east, where they disappeared (Figure 8). This suggests that a decrease in flow velocity occurred from west to east. This also suggests that the

10   top of the sand flat was reworked, with values being higher in the western part due to the bedforms. (Figure 8).





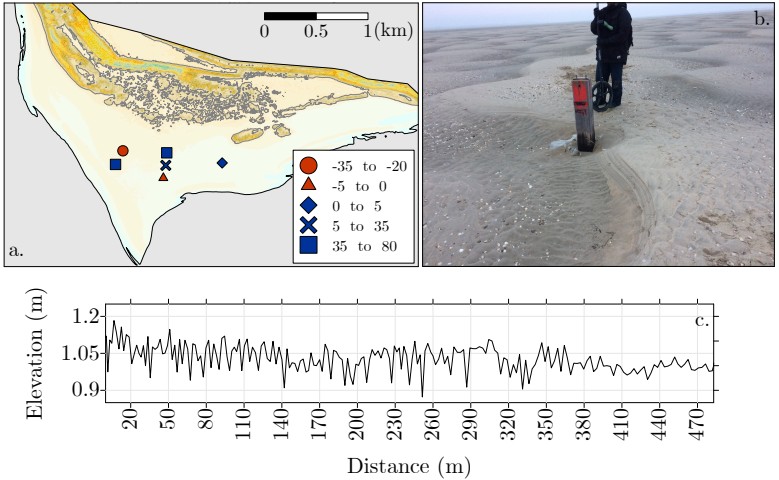

**Figure 8.** a. Results from the survey, showing the change in surface elevation over the storm flooding event, in cm; b. Picture showing and example of bed form developed after the storm at the sand flat; c. Elevation data along the transect shown on Figure 3 showing the bed forms formed during the storm event of January/2017.

## 4.2 Modelled scenarios

Simulation of the storm surge events shows that deposition above the MSHTL happened in almost all tested scenarios. For most scenarios, the sediment is deposited in a clear shore-parallel north-south deposition patch, with volumes varying from $0.7 \ 10^4$ up to $3 \ 10^4 m^3$ of sand. The maximum deposition values occurred for storms j. and k., which are labelled as extreme

storm surges (Figure 9). Only two storms did not yield a significant deposition pattern at the sand flat above MSHTL (storms a. and b., Table 2), with volume values of 474 $m^3$ and 1198 $m^3$ deposited over 2320 and 6203 $m^2$, respectively. These values are distributed in small patches over the plain. Values found in the simulations for the shore-parallel supra-tidal deposition are of the same order of magnitude as the ones derived from the LiDAR data.

Furthermore, simulation results suggest that the amount of sediment deposited tends to be higher for stronger storms. The

amount of deposited sand over the sand flat shows a positive correlation (R>0.8) with hydrodynamic forcing conditions ($Hm0$, $Tp$, $dir$ and $W.L$) (Figure 10). Considering that higher water levels and wave energy are associated with stronger storms, positive values of correlation suggest that stronger storms would lead to more deposition at the sand flat. Even though correlation with main wave direction is also positive, the presence of deposition for all directions suggests that high correlation values are due to the relation between wave energy and wave direction rather than a principal mechanism towards more deposition onto

the sand flat.

We further analyse the relation between hydrodynamic processes and morphological evolution along a cross-shore transect for scenario k analysing the time evolution of 7 parameters: local water level, wave height, cumulative bed level change, bed level change, bed level, convergence values of $u$ in the cross-shore direction (i.e. perpendicular to the shoreline) and zonal components ($u$) of the flow (Figure 11). We also extracted a time-series from a point in the sand flat where deposition occurred





and followed the evolution of water level, wave height, convergence values of $u$ in the cross-shore direction (i.e. perpendicular to the shoreline) and cumulative erosion/accretion (Figure 13).

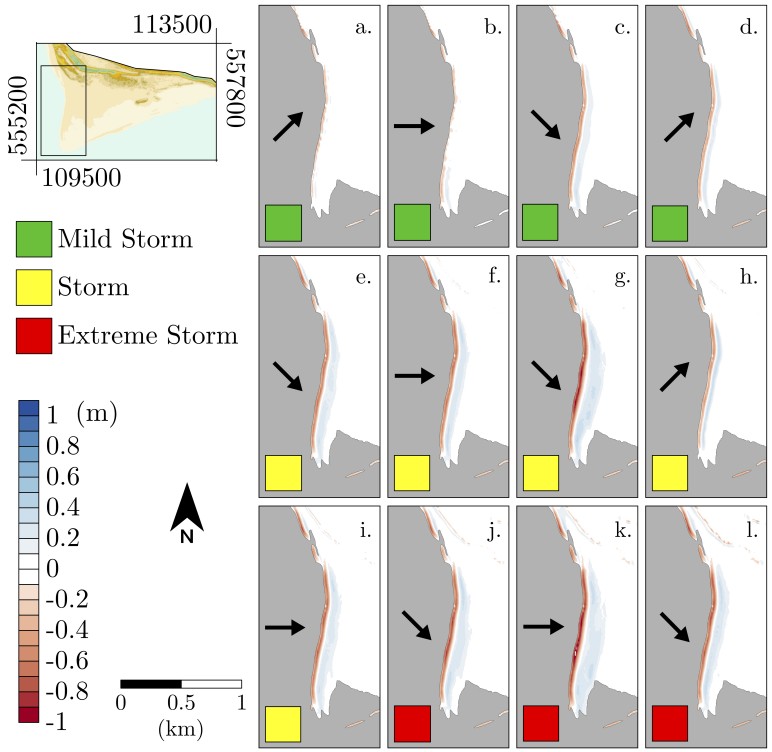

**Figure 9.** Final elevation change after XBeach simulation for all tested scenarios. Arrows represent the main wave direction for the period, whereas the coloured box shows the storm strength.

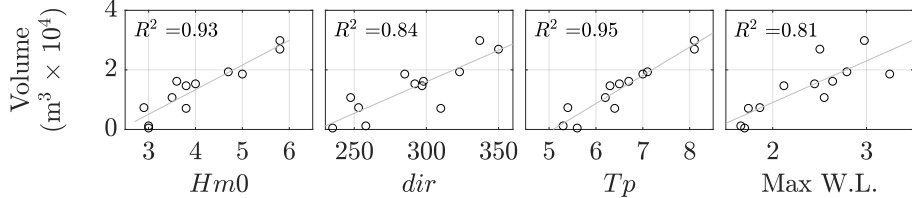

**Figure 10.** Scatter plot for the initial boundary conditions used for each scenario ($Hm0$, $Tp$, $dir$ and maximum water level) against total volume deposited onto the sand flat.

Regarding currents, Figure 11g. shows the cross-shore component of the depth-averaged currents. Before inundation of the sand flat, the system is dominated by an offshore directed current, related to the formation of an undertow current to compensate onshore directed wave-driven mass fluxes. As water inundates the flat, the offshore directed current loses strength,



with a predominance of an onshore-directed current in the upper part of the beach. It is important to notice that as the undertow loses strength, water fluxes in this zone of the beach are less intense compared to water fluxes in elevations above MSHTL.

Most of the deposition occurred at the beginning of the inundation. Using scenario k. as an example, we extracted information from a profile and a point in space, as highlighted in Figure 11 and Figure 13. Results show that deposition occurred mostly

5   between 2:00 and 4:00 hours, which is also the period when water levels reached sufficient elevation to inundate the flat. Between 2:00 and 4:00, values of wave height found in the flat are in the order of 0.1-0.5 meters. After 4:00, the increase of water levels reduces wave dissipation and, in turn, allow wave height in the order of 0.5-0.6 meters on the flat. This suggests that the deposition is a wave-driven process, which may be associated with wave breaking. As the water starts to inundate the flat, wave breaking start to erode the beach. As the breaking evolves as an onshore-directed water flux, it transports the eroded

10  sediment in the down-wave direction. This process occurs for the period in which water depth is small enough to dissipate most of the wave energy, which is supported by the really small waves on the flat. As water depth increase, there is a reduction of wave dissipation, that in turn reduces the sediment transport capacity by either reducing the generated flux or the bottom stress.

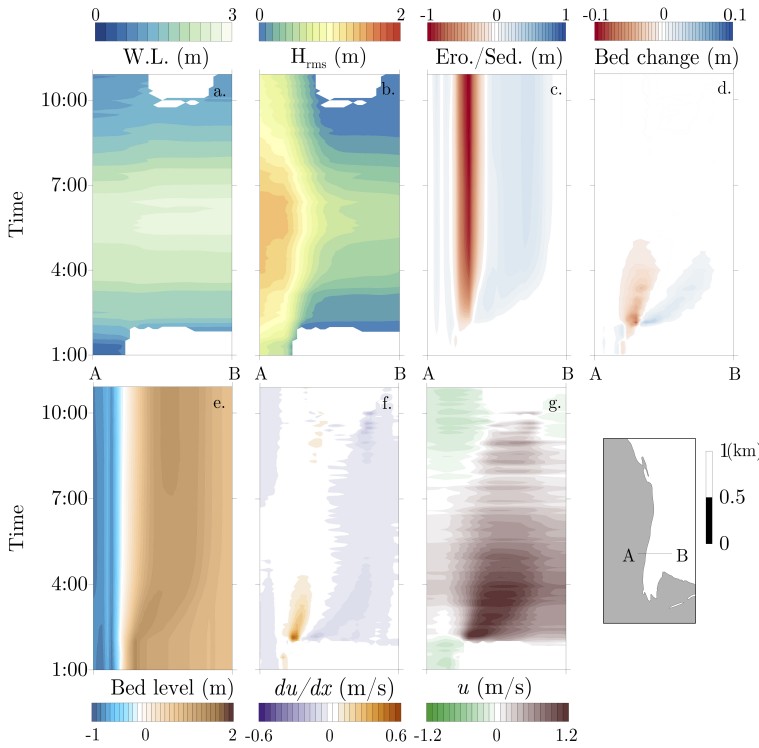

**Figure 11.** Evolution of hydrodynamic and morphological characteristics along the transect A-B taken from scenario k. Y-axis represent the time, whereas the X-axis represent the distance between A and B (left to right), shown on the small reference plot. Variables shown are: local water level relative to NAP (a.), Wave height (b.), cumulative bed level change (c.), bed level change (d.), Bed level (e.), cross-shore convergence of $u$ (f.) and zonal components ($u$) of the flow (g.)





Convergence values of the cross-shore current component ($u$) help to explain the mechanism of deposition further. Positive values of $du/dx$ occur immediately at the beginning of the inundation phase, as water level reaches values of 1.8 meters (Figure 13). Positive values, which relate to the divergence of currents, can be related to an immediate acceleration of water fluxes due to wave breaking in the cross-shore direction. Moreover, the divergence of currents will also lead to erosion of the beach. As

the water starts to inundate the upper part of the beach, wave-driven water fluxes start to decelerate in the cross-shore direction, resulting in a zone of convergence, leading to deposition.

Comparing the amount of sediment eroded on the sand flat and the intertidal zone shows that most sediment is eroded from positions between the MSHTL and the 97.5 quartile (i.e. 1.32 meters) of the daily maximum water level statistics, with two main elevation peaks of erosion located at the extremes of the distribution (Figure 12, SF$i$ and I$i$). In terms of volume,

deposition onto the sand flat from simulations are of the same order of magnitude as those extracted from the LiDAR data. Moreover, the amount eroded from the intertidal zone is one/two orders of magnitude smaller than the portion deposited on the sand flat, suggesting that most of the sediment does not move between intertidal and supra-tidal zones, but is instead reworked within each zone (Figure 12, SF$i$ and I$i$). Moreover, most of the volume eroded from the sand flat refers to areas that became intertidal zones by the end of the simulations, suggesting being areas close to the interface between the intertidal zone (Figure

12, "SF to I"). It is important to note that by comparing volume of deposition on the interdal zone, most of the eroded from the sand flat is deposited in the sand flat itself.


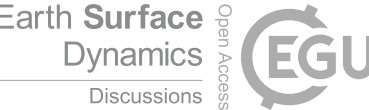


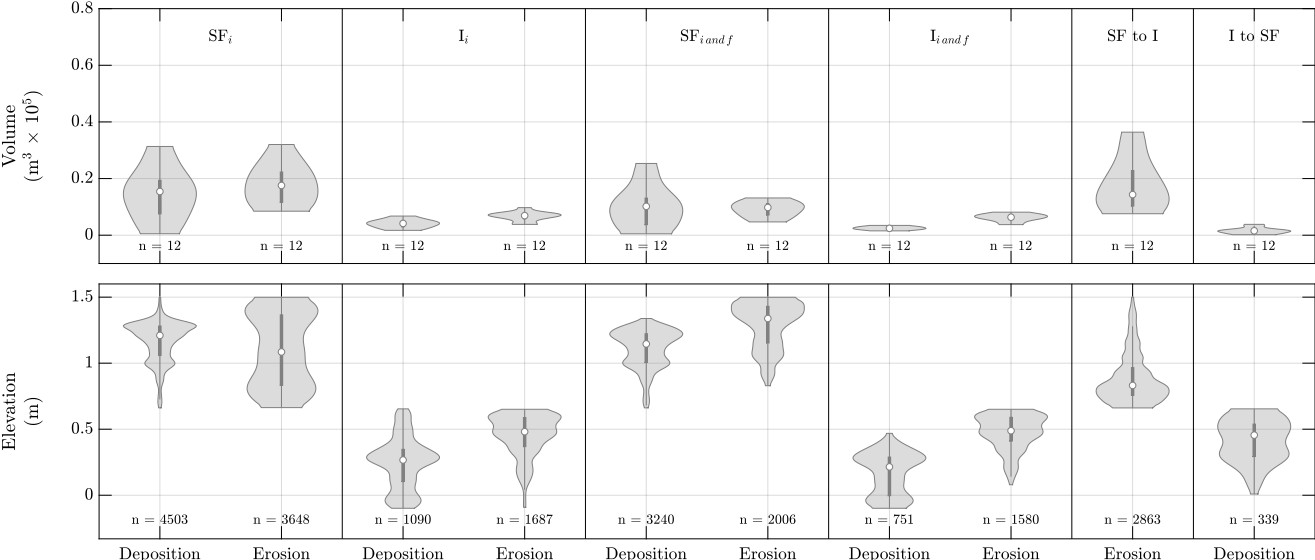

**Figure 12.** Violin plots showing storm-induced volume changes and the elevation where such changes occurred. Dark grey boxes represent a standard boxplot (i.e. median and quartiles) whereas Light grey shapes represent the normalised population distribution of the data plotted in the boxplot. "SF" and "I" refer to "Sand flat" and "Intertidal" zone, respectively. Subscripts "$i$" and "$f$" refer to the boundaries used to estimate each zone that locates the changes. "$i$" refers to a zone (e.g. SF or I) estimated using the initial contour to set the zone (i.e. before the storm), whereas "$f$" refers to the use of the final contour to estimate the zone (i.e. after the storm). Combination of both leads to a restricted zone estimated using both the initial and final contours (thus zones that did not change over the simulation), whereas using both "SF" and "I" refers to zone that changed to the other during after the simulation.

Even though most of the sediment deposited is related to a reworking of the sand flat, some transfer between sub-tidal and supra-tidal still happen. Using the transect A-B from scenario k as an example, results show that median values of elevation where sediment was eroded are 0.61 meters, whereas sediment was deposited on a median elevation of 1.34. Moreover, 85% of the deposition over the whole period occurred in elevations above the MSHTL, whereas 34% of the erosion occurred in elevations above MSHTL. That suggests that sediment was transported from a regularly hydrodynamically active zone (i.e. below MSHTL) to a zone with a sparser occurrence of hydrodynamic processes (i.e. above MSHTL). Moreover, results show that accretion also occurred in areas below 0 meters. This deposition occurs mainly before the inundation of the sand flat and is mainly associated with the offshore-directed current which develops before the inundation phase. Sediment is eroded from the upper beach and transported towards the sea, is then deposited in regions below mean sea level.





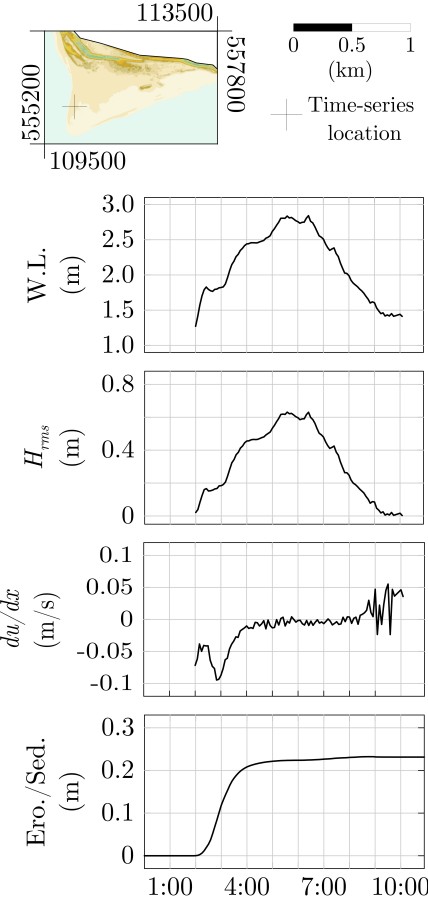

**Figure 13.** Time series of hydrodynamic and morphological characteristics extracted from scenario k.

Using the volume deposited on the sand flat from the simulations, it is possible to estimate the amount of sediment deposited on the sand flat, in reality, using regression techniques. Using both the initial water level and wave height from the simulations as predictors, we could pair the results with measured water levels and wave height in reality. Estimates of sand being deposited on the sand flat show that, between 1997-2017 (i.e. dates which we have LiDAR data and dune volume estimates), the amount

5  of sand predicted to be deposited on the sand flat accounts for 67% of the total sand deposited at the dunes (Figure 15). Curves remain similar up to the year of 2007, where divergence occurs due to a mild period in terms of storms. The unchanging rate of the volume increase in the LiDAR data suggests that even though storm-induced deposition may account for a significant portion of the deposited volume, it is not the only source mechanism of sand for dune growth.



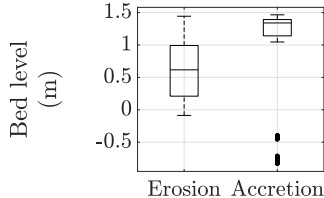

**Figure 14.** Box plot of the elevation where erosion or accretion occurred extracted from scenario k. as example.

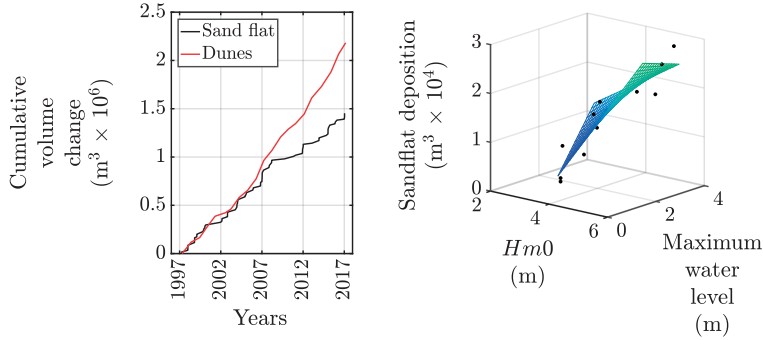

**Figure 15.** Left: Cumulative volume changes from the dunes using LiDAR data (Dunes) and estimated sand flat deposition using regression model as predictor (Sand). The regression model has been built using simulation results with maximum water level at the boundary and wave height as predictors for deposition. Right: Scatter plot of the regression model used with regression surface.

## 5 Discussion.

Overall, both elevation survey data and modelling results suggest that: (i) there is a shore-parallel deposition pattern that occurs at the sand flat in areas above the MSHTL (ii) the deposition can be linked to storm events and (iii) the amount of sediment deposited might have a significant importance for dune growth in the area.

5    As mentioned by Wijnberg et al. (2017), the magnitude of the sand flat surface area is in the order of $2km^2$. Considering the total amount of sand accreted at the dunes and considering the sand flat as its only source, the same amount would represent a lowering in the order of 1 meter in the height of the sand flat. Considering the stability of the sand flat in a yearly scale, which can be seen through the variance and rate of change maps at the sand flat, Wijnberg et al. (2017) suggests that either the sand flat has been continuously replenished by sand or that the supra-tidal part of the sand flat is not the primary source of

10   sediment for the dunes. Our elevation survey data results suggest that the accretion above MSHTL can contribute with more than 27% of the sediment supply of the dunes on a yearly basis. Furthermore, numerical modelling results support that storms may act as a depositional mechanism onto sand flats, depositing similar shore-parallel supra-tidal deposits of sand as seen in





the elevation data. Also, estimates pairing modelling results and actual data also suggest that cumulative depositions would be of the same order of magnitude of volume changes at the dunes. The potential to contribute to more than half of the yearly average deposited volume in the dunes suggests that, for a sand flat setting like Texel, the sediment deposited through storm surge flooding can be seen as an important mechanism in terms of dune growth.

However, surprisingly, numerical modelling also shown that the such shore-parallel supra-tidal deposits are not from the sub-tidal zone as previously hypothesized, but is rather a product of reworking of mostly supra-tidal deposited sand close to the intertidal zone. Considering that the sand reworked into a depositional ridge is already from the supra-tidal zone (thus, potentially available for transport), the question arises whether this mechanism can be genuinely considered a new source for dune growth. To be considered as a new source, the supra-tidal zone of the sand flat should not act as a primary source,

as hypothesized by Wijnberg et al. (2017). Although the elevation stability of most of the supra-tidal zone of the sand flat, together with no growth trend related to the depositional ridge, hints at such possibility, the main process that might lead to such limitation of the zone to act as a sediment source remains unclear. Silva et al. (2018) shows that spatial variation in groundwater depth along the sand flat induces variations in sediment supply to the dunes, which may lead to an overall limitation of the sand flat to act as a source in the long-term. Hoonhout and de Vries (2017) suggested that in a mega-nourishment setting, wind

transport would lead to a sorting process of the sediments at the beach surface that, within a certain period of time, would induce an armouring effect that could reduce the potential of the sediment surface to act as a sediment source. Considering that large parts of the sand flat remain exposed most of the time, there is a possibility of armouring effects reducing the capacity of the flat to serve as a sediment source. Although possible, to what extent the armouring effect does also occur in a sand flat setting and what effect this has on the sediment transport towards the dunes remains for further research.

On coastlines away from inlets, storm surges tend to reach the dune toe, erode the sediment and transport it towards the sub-tidal zone via a strong undertow current (Aagaard, 2014; Guannel and Özkan Haller, 2014). For sand flats, this mechanism holds at the start of the event, before sufficient levels for inundation. After inundation, the undertow weakens, and sediment eroded from the upper beach is transported onto the sand flat instead of seaward, being deposited mainly by deceleration of a wave-driven flow, creating a shore-parallel depositional ridge onto the sand flats on areas that are not reworked by the sea in a

daily basis. The fresh reworked and deposited sand further inland tend to be less affected by surface moisture variations induced by tide movements and, therefore, are more prone to synchronise with energetic wind events that are capable of transferring this sand to the dunes.

Although modelling results suggest that the amount of sand deposited is directly proportional to storm strength (i.e. storm surge level plus wave energy), data analysis does not show statistical evidence that it happens in reality. This discrepancy may

be explained by: the annual time interval of the surveys; cumulative effect of multiple storms before the total dispersion of the deposition of the previous one; the date which the measurement has been taken, since surveys done close to storms would have a higher probability of picturing the shore-parallel deposition pattern; changes in the sand flat shape between storms, which might lead to slightly non-uniform hydrodynamic forcing in time, thus influencing the potential capacity of sand to be transferred from the sub-tidal to the supra-tidal zone.





Currently, it is hard to quantify exactly how much sand related to storm deposition or remobilisation of previously deposited sand contributes to dune growth. LiDAR results show that the sediment deposited on the sand flat represents more than a quarter of the sediment necessary to maintain the dune increase at the rates that have been measured. However, being available for transport does not mean that the sediment will indeed end up at the dunes, since other hydrodynamic processes (e.g. next

storm surge, erosion due to channel migration) may transport it to the sub-tidal area. Furthermore, the depositional ridge developed between LiDAR data has been already reworked by other wind and potentially other storms due to the time between the measurements. Moreover, wind can also transport this sand back to the sub-tidal zone depending on its direction. Furthermore, limiting factors such as surface moisture and lag deposits can reduce the capacity of the wind considerably to transport the sand from the sand flat towards the dunes (Delgado-Fernandez and Davidson-Arnott, 2011; de Vries et al., 2012; Bauer et al.,

2009; Houser and Ellis, 2013; Duarte-Campos et al., 2018). Thus, synchronisation of capable wind events, bed/grain characteristics and available sediment plays a key role (Houser, 2009). Nevertheless, considering the capacity of sediment deposition suggested by our results together with the dominant wind direction, it is probable that at least part of this sediment contributes to dune growth.

Considering that the depositional ridge on the sand flat is mainly composed by reworked supra-tidal deposits, it remains

unclear which mechanisms are responsible for sediment exchange between supra-tidal and sub-tidal zones on sand flats. As a hypothetical framework of sediment exchange, we propose four phases of sediment pathways. Initially, longshore transport would transport sediments from northern coastlines of the island and deposit on the sub-tidal and intertidal exposed coastline of the sand flat. Local processes such as longshore gradients would shape the shoreline creating locations of accretion and erosion. On accretive portions, fresh sediment would become available on the supra-tidal zone through local progradation and

accommodation of sediment previously located at the intertidal zone. Swash and overwash processes would be responsible for exchanging sediment between subtidal and intertidal zones. During storm surges, such fresh sediment close to MSHTL would be reworked into the supra-tidal ridges. Finally, aeolian transport would be responsible for eroding such depositional ridge. Berm formation and supra-tidal shore-parallel depositional ridges on open coastal beaches have been already related to deposition of sediment related to swash processes (Houser and Ellis, 2013). Several authors exemplify that exchange of

sediment between sub-tidal and supra-tidal zones depend on surf and swash processes during calm conditions or migration of sub-tidal and intertidal bars landward (Houser and Ellis, 2013; Aagaard et al., 2004; Jackson et al., 2004; Houser and Barrett, 2010). Our present research suggests that, for a sand flat setting, a supra-tidal shore-parallel depositional ridge can also form during a storm surge flooding, inducing the deposition of a certain amount of sand that is in the same order of magnitude of the amount of dune volume increase.

**6 Conclusions**

A sand flat at the tip of the island of Texel (NL) has been used as a case study to firstly identify processes and storm properties that cause deposition on the sand flat during storm-surge flooding and discuss the relationship between the supra-tidal deposition and sand supply to the dunes. The case was approached by an integrated analysis of LiDAR surveys, field survey



and numerical modelling. Results suggest that supra-tidal deposition of sand is directly proportional to storm surge levels and wave energy. Also, the amount of sand deposited may account for more than a quarter of the volume deposited at the dunes in a yearly basis. During storm surge flooding of the sand flat, sediment is mostly eroded from supra-tidal areas close to MSHTL and deposited further landward by a wave-driven onshore directed flow. Furthermore, simulation results suggest that most of
5 the deposition occurs at the beginning of the sand flat inundation. Deposition is controlled by the convergence of the cross-shore component of the wave-driven flow. Furthermore, storms are also capable of remobilising the top layer of sediment of the sand flat, making fresh sediment available for aeolian transport if an armouring effect occurred, especially in the west part of the sand flat. Therefore, in a sand flat setting, storm surges have the potential of reworking considerable amounts of sand into depositional ridges that are further transported by the wind to the dunes. This suggests that storms play a significant role
in supplying sand for the dunes to grow in a sand flat setting.

*Code and data availability.* Data used related to long-term Dutch monitoring program can be found at *opendap.deltares.nl*. Remaining datasets and code are available upon request from Filipe Galiforni-Silva (f.galifornisilva@utwente.nl).

*Competing interests.* The authors declare that they have no conflict of interest

*Acknowledgements.* This research forms a component of the CoCoChannel project (Co-designing Coasts using natural Channel-shoal dy-
15 namics), which is funded by Netherlands Organization for Scientific Research, Earth Sciences division (NWO-ALW), and co-funded by Hoogheemraadschap Hollands Noorderkwartier. We further wish to acknowledge Rijkswaterstaat and KNMI for making their valuable bathymetric and topographic data sets freely available, as well as the water level, wave and wind data. Furthermore, we would like to thank Prof. Dr Dano Roelvink for his help and discussions on the application of the XBeach model, and Dr Theo Gerkema for the data used in the model validation.



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
