# Peer review of "Storm-induced sediment supply to coastal dunes on sand flats"

_Earth Surface Dynamics, 2019_

## Referee Comment (RC1) · Anonymous Referee #1 · 21 Feb 2020

This paper investigates sand supply to coastal dunes in an inlet setting using both field surveys, Lidar data, and numerical modelling using Xbeach. The authors conclude that a significant amount of sand deposited in the dunes is supplied from the sand flat fronting the dunes, which is a reasonable suggestion. The manuscript has been sub-stantially improved from the first version. I do have, however, a few specific comments that the authors might want to consider.

Specific comments

The Xbeach model runs comprise 12 surge events but only one bathymetry (surveyed in 2009) is used. Would this introduce error in the simulations? Do you have any idea? You might want to address and discuss whether or not changing bathymetries might affect you results. I am also somewhat surprised that you use ADCP data from the tidal

inlet to validate the simulations over the sand flat. The inlet is a totally different environment with different processes (tidal currents vs wave-driven currents) being dominant. Moreover, the fit between the simulated and measured current speeds (Figure 4) is not great. I doubt that the validation exercise is relevant for conditions at the tidal flat and I actually suggest you leave out the validation and just go with the fact that you use the model in default mode.

Sediment deposition at the sand flat is caused by 'an onshore-directed water flux' (p.15, l.9). Could you be more specific? What drives this current? Is it homogeneous in the vertical? Are the simulated current speeds averages over the water column, or is there vertical segregation? Why is the current onshore when the flat is inundated?

The sediment deposition is calculated through regression techniques (p.18). Could you tell us a bit more about how you actually calculate those volumes? Do you use the scatterplots in Figure 10? In the discussion section, you compare the volume increase in the dunes with the volume increase from annual Lidar data, and with the results of numerical simulations of storm events. You conclude that between 27% and 67% of the sand added to the dunes come from the sand flat. Where does the rest come from, then? I do not think you can make this direct comparison. The annual Lidar surveys would miss all the sediment being bypassed across the sand flat during the period between two surveys, and the simulations comprise only surge situations with non-storm events left out. In short, I believe all the sand comes from the subtidal and is subsequently transported across the sand flat. If the sand flat were not replenished from offshore/longshore sources, it would have disappeared a long time ago.

Technical corrections

p.4, lines 8 & 10: 'storms surges' should be 'storm surges'.

p.5, l.1-3: The dune area has been defined as the area lying above the +3 m contour and the sand flat is located between +1.5 m and the MSHTL (earlier defined as +0.66 m). What lies in between?

p.9, l.3: '....deposition patterns in the supra-tidal zone occur between at least ten different years'. What does this passage mean?

Figure 7: What do the black and the red boxes represent?

p.16, l.15: 'interdal' should be 'intertidal'.

---

## Referee Comment (RC2) · Edward Anthony (Referee) · 27 Feb 2020

Storm-induced sediment supply to coastal dunes on sand flats Filipe Galiforni-Silva, Kathelijne M. Wijnberg, and Suzanne J.M.H. Hulscher

The manuscript submitted by Galiforni-Silva et al. on storm-induced sediment supply to coastal dunes on sand flats is a well-crafted piece of research that clearly shows how storm surges can contribute to the build-up of dunes where sand flats provide a surface for aeolian supply. The paper complements our understanding of the range of processes - in this instance storm surges - involved in the construction of coastal dunes. The manuscript does not have any flaws that should prohibit its publication. The data and methodology are pertinent to the aims of the study and clearly produced

results that substantiate these aims. The methods used by the authors are sufficiently detailed and transparent to enable reproducibility. The combination of field observations and modelling enables a better apprehension of patterns of multi-decadal sedimentation. The data interpretations and conclusions are robust, viable and reliable. In terms of applicability of these findings elsewhere, the results confirm the processes of storm surge-induced sedimentation that led to the build-up of an anomalous dune, now well abandoned inland in Ghyvelde, isolated within a sand-flat environment near a former tidal inlet in the Flemish Coastal Plain, further south of the present Texel Island site (See reference 1 below). The manuscript has given proper attention to previous references on the subject and study area.

Minor comment: Regarding the armouring effects discussed on page 20, could there be a salt-crusting effect too that contributes to armouring? (See reference 2 below)

Minor typo/grammatical points and accords that I have corrected: • Page 2, line 2: Âă. . ..for a tide-dominated beach on the coast of France,. . . • Page 4, lines 7-14: . . ... Storm surges with maximum water levels between the 75 quartile (0.86 meters) and 1.9 meters above MSL are classified as mild, whereas maximum water levels between 1.9 and 2.6 meter above MSL are classified as normal storm surges, and above 2.6 meters as extreme storm surges. To determine the local storm climate, we used a 29-year long time series of hourly water levels collected at the 'Den Helder' tide gauge. . . • Page 6, Line 3: Âă. . . areas where more accretive or erosive events occurred, regardless of their magnitude and trends over time. • Page 8, Line 8: Âă. . .. of possible storms for validation based only on periods of mild storm surges. • Page 9: Âă4 Results. 4.1 Supra-tidal development Figure 5 presents elevation difference maps over consecutive years. Maps show that deposition patterns in the supra-tidal zone occur over at least ten different years. For some years, as between 1998-1999 (b.) and 2003-2004 (e.), the deposition extended from the north to the south of the flat, and occurred at least 100 meters landward of the mean spring high tide level (i.e. higher elevations). For other periods, such as j., l. and r., the deposition occurred much closer

to the MSHTL, although also oriented from north to south. For others, such as m, the deposition occurred only in the southern tip of the flat. Erosion patterns exceeding 0.16 meters occurred only over a few years,. . . • Page 15, Line 9: Âă. . .. flat, breaking waves start to erode the beach. • Page 15, Line11: Âă. . . As water depth increases, there • Page 17, Line 2: Âă. . . supra-tidal still occurred. Using. . . • Page 20, Line 5-6: ÂăÂăHowever, surprisingly, numerical modelling has also shown that such shore-parallel supra-tidal deposits are not from the sub-tidal zone as previously hypothesized, but are rather a product of.. • Page 21, Line 6: . . .Âăreworked by wind and potentially other storms. . .

- Reference on inland dune development from storm-surge deposition over sand-flat: Anthony, E.J., Mrani-Alaoui, M., Héquette, A., 2010. Shoreface sand supply and mid- to late Holocene aeolian dune formation on the storm-dominated macrotidal coast of the southern North Sea. Marine Geology, 276, 100-104. https://doi.org/10.1016/j.margeo.2012.01.001

- Reference on salt-crusting and impact on dune grain stability Langston, G., McKenna Neumann, C., 2005. An experimental study on the susceptibility of crusted surfaces to wind erosion: A comparison of the strength properties of biotic and salt crusts. Geomorphology, 72, 40-53. https://doi.org/10.1016/j.geomorph.2005.05.003

---

## Author Comment (AC1) · 4 Mar 2020

Dear reviewer, Thank you very much for your valuable comments and time when evaluating our work. Your feedback truly helped our manuscript, highlighting essential aspects that were not clear in the previous version. You can find our specific comments below. Comments that need editing in the text will be acknowledged here but addressed in the revised version.

Specific comments

- The Xbeach model runs comprise 12 surge events but only one bathymetry (surveyed in 2009) is used. Would this introduce error in the simulations? Do you have any idea? You might want to address and discuss whether or not changing bathymetries might

affect you results.

R: The idea behind using the same bathymetry was to analyze only the effects of the water level and the waves onto the sand flat. With that said, we think that changing the bathymetry may lead to different results quantitatively, but not necessarily qualitatively. The process leading to sediment deposition seems to be independent of the bathymetry since it is mainly driven by the inundation and the consequent on-shore wave-driven current that develops at the start of the surge. Moreover, LiDAR data shows that the deposition onto the sand flat occurred in different bathymetric settings. However, we do acknowledge that, with different bathymetries, local changes may lead to more or less sediment being transported/deposited.

- I am also somewhat surprised that you use ADCP data from the tidal inlet to validate the simulations over the sand flat. The inlet is a totally different environment with different processes (tidal currents vs wave-driven currents) being dominant. Moreover, the fit between the simulated and measured current speeds (Figure 4) is not great. I doubt that the validation exercise is relevant for conditions at the tidal flat and I actually suggest you leave out the validation and just go with the fact that you use the model in default mode.

R: We agree with everything, as briefly stated in validation subsection in the previous version. We take the reviewer suggestion, and the validation section will be removed in the revised version. We will maintain a short paragraph explaining our limitations regarding validation and our choices on using the model as default.

- Sediment deposition at the sand flat is caused by 'an onshore-directed water flux' (p.15, l.9). Could you be more specific? What drives this current? Is it homogeneous in the vertical? Are the simulated current speeds averages over the water column, or is there vertical segregation? Why is the current onshore when the flat is inundated?

R: Indeed, the current text needed more details, which will be added in the revised version. The model is depth-averaged, though it includes known processes that induce

vertical segregation of the flow (i.e. undertow). The way we see the process is as follows: At the start of the event (i.e. before sand flat inundation), wave breaking induces pressure gradients that are higher at the shoreline, causing the development of the undertow and a depth-averaged offshore barotropic flow. As the water level increases and the sand flat gets inundated, the pressure gradient induced by the surf bore reduces, as it starts to flow over and enter the sand flat. That leads to a reduction of the offshore flow, and the surf bore starts to lead the net flow component, resulting in an onshore-directed flow. It is important to note that currents driven by gradients on the momentum flow, which usually leads the development of other types of nearshore circulation (e.g. longshore currents) did not lead to any statistically significant relationship with the deposition.

- The sediment deposition is calculated through regression techniques (p.18). Could you tell us a bit more about how you actually calculate those volumes? Do you use the scatterplots in Figure 10?

R: For this part of the text, specifically, we estimate the potential amount of sand deposited using time-series of water level and waves as predictors, as shown in Figure 15. As we have seen a good agreement between both in Figure 10, we attempted such a quantification of the volume of sand. With that said, it is important to note that, in reality, several other variables may influence such a deposition. Moreover, as our model is not validated for this specific site, uncertainties around quantitative estimates arise.

- In the discussion section, you compare the volume increase in the dunes with the volume increase from annual Lidar data, and with the results of numerical simulations of storm events. You conclude that between 27% and 67% of the sand added to the dunes come from the sand flat. Where does the rest come from, then? I do not think you can make this direct comparison. The annual Lidar surveys would miss all the sediment being bypassed across the sand flat during the period between two surveys, and the simulations comprise only surge situations with non-storm events left out. In

short, I believe all the sand comes from the subtidal and is subsequently transported across the sand flat. If the sand flat were not replenished from offshore/longshore sources, it would have disappeared a long time ago.

R: Indeed, we also believe that all the sediment comes from the subtidal part depending on the time-scale. It has also been framed in the discussion (page 11, lines 1-30). We believe that the remaining sand would come from the intertidal zone during mild periods, after being transported and deposited in this zone by onshore cross-shore sediment transport from the subtidal zone. The point that we try to make is that storm surges would facilitate this process by either remobilizing the top layer of the sand flat and also by depositing a significant amount of sand in an always exposed area. We completely agree that there is a lot of missing information due to the time-resolution of our LiDAR data. However, the comparison is only to show the order of magnitude that we refer when talking about the deposited sand onto the sand flats.

Technical corrections

- p.4, lines 8 & 10: 'storms surges' should be 'storm surges'.

R: We will correct in the revised version.

- p.5, l.1-3: The dune area has been defined as the area lying above the +3 m contour and the sand flat is located between +1.5 m and the MSHTL (earlier defined as +0.66 m). What lies in between?

R: We limit the extent of the sand flat up to 1.5 meters since above such an elevation there are already signals of aeolian deposition and incipient dune formation, which would add an error on the hydrodynamic deposition estimates. We will add a further explanation in the revised version.

- p.9, l.3: . . ..deposition patterns in the supra-tidal zone occur between at least ten different years'. What does this passage mean?

R: Will be rephrased for clarity.

- Figure 7: What do the black and the red boxes represent?

R: Colours represent accretion and erosion. Further information will be added to the Figure caption.

- p.16, l.15: 'interdal' should be 'intertidal'.

R: We will correct in the revised version.

---

## Author Comment (AC2) · 4 Mar 2020

Dear Prof. Dr. Edward Anthony,

Thank you very much for your time to review our manuscript. It was with much enthusiasm that we received such nice comments on our paper coming from a well-known expert in the theme. We have carefully revised the manuscript accordingly, and we give the responses to each comment below. Minor typo/grammatical points will be corrected in the revised version.

Minor comments

- Regarding the armouring effects discussed on page 20, could therebe a salt-crusting effect too that contributes to armouring? (See reference 2 below)

[Figure]

R: Indeed, salt-crusting may be a potential effect that contributes to the armouring. Of course, a definite idea of which process leads the armouring would require a dedicated study in the area. However, we believe that, regardless of the leading processes, the reworking of the top layer due to the storm surge would be enough to reduce the armouring and enhance supply to the dunes. The reference of this process will be added in the discussion part of the revised version.